# Vaccines Alone Cannot Slow the Evolution of SARS-CoV-2

**DOI:** 10.3390/vaccines11040853

**Published:** 2023-04-16

**Authors:** Debra Van Egeren, Madison Stoddard, Laura F. White, Natasha S. Hochberg, Michael S. Rogers, Bruce Zetter, Diane Joseph-McCarthy, Arijit Chakravarty

**Affiliations:** 1Department of Medicine, Weill Cornell Medicine, New York, NY 10021, USA; 2New York Genome Center, New York, NY 10013, USA; 3Fractal Therapeutics, Lexington, MA 02420, USA; 4Department of Biostatistics, Boston University School of Public Health, Boston, MA 02118, USA; 5Novartis Institutes for Biomedical Research, Cambridge, MA 02139, USA; 6Department of Medicine, Boston University School of Medicine, Boston, MA 02118, USA; 7Department of Surgery, Harvard Medical School, Boston, MA 02115, USA; 8Vascular Biology Program, Boston Children’s Hospital, Boston, MA 02115, USA; 9Department of Biomedical Engineering, Boston University, Boston, MA 02215, USA

**Keywords:** SARS-CoV-2, evolution, viral variants, mathematical modeling, immune evasion

## Abstract

The rapid emergence of immune-evading viral variants of SARS-CoV-2 calls into question the practicality of a vaccine-only public-health strategy for managing the ongoing COVID-19 pandemic. It has been suggested that widespread vaccination is necessary to prevent the emergence of future immune-evading mutants. Here, we examined that proposition using stochastic computational models of viral transmission and mutation. Specifically, we looked at the likelihood of emergence of immune escape variants requiring multiple mutations and the impact of vaccination on this process. Our results suggest that the transmission rate of intermediate SARS-CoV-2 mutants will impact the rate at which novel immune-evading variants appear. While vaccination can lower the rate at which new variants appear, other interventions that reduce transmission can also have the same effect. Crucially, relying solely on widespread and repeated vaccination (vaccinating the entire population multiple times a year) is not sufficient to prevent the emergence of novel immune-evading strains, if transmission rates remain high within the population. Thus, vaccines alone are incapable of slowing the pace of evolution of immune evasion, and vaccinal protection against severe and fatal outcomes for COVID-19 patients is therefore not assured.

## 1. Introduction

As the COVID-19 pandemic continues to rage worldwide, it is generally hypothesized that the next phase of the crisis will involve a widely circulating disease with limited virulence [1,2,3]. This belief, often articulated as “learning to live with COVID” [4,5], assumes that vaccines can be used to keep the fatality rate of COVID-19 infections in check even in the face of high levels of viral transmission [6]. Over the past year, however, the rapid emergence of immune-evading viral variants of SARS-CoV-2 [7] has cast a pall over this vision of the future [8,9].

With the rapid worldwide spread of the Omicron variants, public-health authorities have stepped up their efforts to increase vaccination rates, with a particular focus on boosters [10,11]. Two main assumptions underpin the narrow focus on vaccination as a strategy for managing COVID-19 disease burden, despite the demonstrated frequency of breakthrough infections. The first assumption is that vaccines will continue to protect against severe outcomes (hospitalization and death), and the second is that vaccines will partially reduce transmission. The first assumption appears to remain valid, as vaccination appears to reduce the risk of severe disease and hospitalization [12,13,14], although this effect wanes quickly without additional booster doses [13] and is vulnerable to viral evolution [14]. Characterization of breakthrough infections has led to uncertainty around the second assumption, as vaccinal protection against SARS-CoV-2 infection has waned rapidly with the emergence of new viral variants [15,16,17]. Vaccine breakthrough cases present similar levels of peak viral load [18,19], although this viral load appears to wane more rapidly than in unvaccinated infections [19,20,21]. Notably, the household secondary attack rate is modestly reduced (by 30–50%) in vaccine breakthrough index cases relative to the unvaccinated [15,22,23].

At this point in time, a case can be made that a crucial determinant of the eventual resolution of the crisis phase of the pandemic will hinge on the ongoing rate of viral evolution. Each new variant brings with it a set of unknowns in terms of key viral properties—the transmissibility, the degree of immune evasion, the superspreading, and infection fatality rate (see Table S3 in [24]). A scenario involving repeated waves of infection driven by the emergence of new variants thus poses tremendous challenges for formulating and communicating public-health strategies, as has been observed already for previous waves of the pandemic [25,26]. The rate of evolution of viral immune evasion also threatens to degrade the utility of vaccines as part of the public-health response to COVID-19. Viral immune evasion holds the potential to reduce vaccinal efficacy against infection (and hence transmission) and raises the possibility of reductions in vaccinal protection against severe disease outcomes. These reductions in vaccinal efficacy further threaten to hinder vaccinal uptake by providing ammunition to vaccine skeptics. As a consequence, slowing the rate of viral evolution is recognized as a crucial objective for managing the current pandemic [10,27,28,29].

In this context, some reports have emphasized that widespread vaccination is critical for slowing viral evolution [10,30,31]. The connection between vaccine uptake and slowing the pace of viral evolution is not intuitive and may rely on assumptions about the impact of vaccines on transmission that have changed as the pandemic has evolved. Alternatively, it may arise from conflating vaccinal impact on severe disease with vaccinal impact on infection. In either case, the link between vaccine uptake and viral evolution bears closer examination, as it has not been a significant area of study thus far.

We thus sought to examine the impact of a public-health strategy that relies solely on vaccine uptake on the rate of viral evolution. To do this, we created a stochastic computational model to investigate the emergence of new immune escape variants with multiple mutations to answer the question “what factors determine the rate of emergence of novel immune-evading variants?”. We then used this model to study the impact of vaccination on the emergence of new variants, answering the question “what rate of vaccination would be required to blunt this rate of emergence?”.

## 2. Materials and Methods

### 2.1. Branching Process Model of Stochastic SARS-CoV-2 Transmission Dynamics

We simulated the emergence of new SARS-CoV-2 variants with one or more mutations as a stochastic branching process with discrete generations. In this model, each infected individual infects a Poisson-distributed number of new hosts, before they recover. The mean number of infections generated by a single infected individual is the transmission rate *R_T_*. The simulation starts with one individual infected by a SARS-CoV-2 variant with a single mutant, and each transmission event has a 1 × 10^−5^ probability of creating a new variant with an additional mutation. This mutation rate was estimated from a ~1 × 10^−3^/yr SARS-CoV-2 evolutionary rate inferred from phylogenetic analyses [32], which corresponds to a per-site per-transmission rate of 3.8 × 10^−5^ given an infection period of two weeks. These mutations can affect the transmissibility of the virus, so that the *R_T_* of the new variant is larger or smaller than the parent variant. The simulation continues, until there are no more infections that originate from the initial single mutant (i.e., the SARS-CoV-2 population derived from the single mutant goes extinct) or a variant with the target number of mutations is created and survives indefinitely. The probability that a variant with *R_T_* > 1 will survive indefinitely is given by the branching process theory as the minimum solution *π* to $\pi =1-e^{-R_{T}\pi }$ in the interval [0, 1] [33].

Immunity due to previous infection or vaccination lowers the transmissibility *R_T_* of SARS-CoV-2 by reducing the number of uninfected contacts to which an infected individual can successfully transmit the virus. Disregarding demographic and geographic effects on interpersonal contacts and assuming individuals are well-mixed with regards to vaccine or immunity status, the immunity-adjusted *R_T_*’ is $R_{T}\left(1-f_{v}\varepsilon_{v}-f_{r}\left(1-f_{v}\right)\right)$, where *R_T_* is the base transmissibility, assuming no immunity, *f_r_* and *f_v_* are the fractions of individuals who are immune after recovering from prior infection and vaccinated, respectively, and *ε_v_* is the efficacy of vaccination in preventing infection. Here, we assumed that prior infection confers perfect immunity against subsequent infections.

For the initial set of simulations, we assumed that the viral transmission rate *R_T_* is constant over time. We also simulated situations where viral transmission rates decrease over time, to model a wave of SARS-CoV-2 transmission that eventually wanes due to herd immunity or the implementation of interventions such as social distancing. For these simulations, we assumed that the transmission rates for all variants in the population decrease exponentially over time at rate *δ*. Therefore, the time-dependent transmission rate is $R_{T}e^{-\delta t}$, where *t* is the time elapsed since the beginning of the simulation, when the first single mutant appears in the population.

### 2.2. Extended SIR Model of Viral Transmission, Mutation, and Vaccination

To determine how effectively vaccination can suppress the creation of new SARS-CoV-2 variants, we adapted an extended susceptible-infected-recovered (SIR) model of viral transmission and immunity from our previous work, which includes a description of the system of differential equations used to simulate the model [34]. Briefly, this model has two viral variants—the single-mutant variant, which is initially present at a very low frequency (1 in 300 million individuals), and the double mutant, which is not initially present in the population. The single mutant circulates with base *R_T_* ranging from 1 to 5.7 and is countered by a vaccine with a 70% or 100% efficacy against transmission. Each infected individual has a rate of 1 × 10^−5^ per day of generating a new double-mutant infection (explanation for this rate was given in Section 2.1). The population initially has no natural or vaccinal immunity against the new single-mutant variant, and the vaccination campaign against the variant begins just when the new variant appears, where a constant fraction of the unvaccinated susceptible population is vaccinated per day.

### 2.3. Simulation of Intrahost Mutation Dynamics

We adapted our published computational model [35] of SARS-CoV-2 replication, mutation, and selection within hosts to incorporate the observed differences in viral load kinetics between vaccinated and unvaccinated individuals infected with the Delta SARS-CoV-2 variant. Using the estimated values for the peak viral load, the rate of viral load growth and decline, and the time to reach the peak viral load [21], we constructed a viral load curve over time for vaccinated and unvaccinated individuals, assuming there was a total of 10 mL of liquid in the airway of each individual. Using the model from [35] with the default parameter values, we simulated typical-length COVID-19 infections using these estimated viral load curves. To simulate longer infections, we assumed that patients with prolonged COVID-19 remained at the peak viral load for an extended period of time (up to 15 weeks), with the same viral load growth and decline kinetics as in typical-length infections. Each condition was simulated 1000 times to compute the frequency of novel variant emergence.

## 3. Results

### 3.1. The Risk of Generating New SARS-CoV-2 Variants Strongly Depends on the Transmission Rate of Existing Variants

To determine how quickly new SARS-CoV-2 variants with multiple mutations that could evade vaccinal immunity are created, we built a stochastic computational model of viral transmission and mutation. In this branching process model, each infected individual spreads the virus to *X* new hosts after a fixed amount of time (here, two weeks). Here, *X* is a Poisson-distributed random variable with mean *R_T_*, corresponding to the current SARS-CoV-2 reproduction number. Using this framework, we simulated the multistep creation of a new variant with more than one point mutation, which mirrors the process that generates new immune-evading variants [36,37]. At the start of each simulation, a new point mutation occurs in an infected host. This intermediate single mutant is transmitted with reproduction number *R_T_*, until the lineage goes extinct. We assumed new mutations occur in this intermediate single-mutant lineage at a rate of 10^−5^ per transmission event (Section 2). To assess the rate at which new variants emerge, we estimated the probability that a specific new single-mutant lineage would eventually give rise to a new more-transmissible SARS-CoV-2 variant with additional mutations.

Our simulation results showed that the probability that a single-mutant lineage will generate variants with two or more point mutations strongly depends on the reproduction number of the intermediate single mutant (Figure 1). At *R_T_* values below 1, the single mutant will always eventually go extinct, so any additional mutations that occur in the lineage must occur relatively quickly. The probability that a second mutation will occur before the lineage goes extinct increases dramatically if *R_T_* of the single mutant is above 1, since in that case the single-mutant lineage is more likely to survive longer and produce more infections (Figure 1A). The time at which new variants arise also depends on *R_T_* of the intermediate. In cases when a double mutant eventually emerges from the original single-mutant lineage, that double mutant must emerge quickly in cases where *R_T_* is less than 1 (Figure 1B). However, when the intermediate can survive for longer periods of time (*R_T_* > 1), increasing the transmission rate decreases the amount of time necessary for generation of the double mutant (Figure 1B). Additionally, the transmission advantage of the final double-mutant variant does not have much of an impact on the rate at which new double mutants emerge (Figure 1A,B).

If the single-mutant lineage survives long enough to generate a double mutant, it is likely to also be capable of generating variants with more than two mutations. As the number of mutations in the final variant increases, the eventual probability of creating the final variant remains fairly constant (Figure 1C). However, variants with more mutations require more time to develop (Figure 1D).

### 3.2. Reducing the Transmission Rate through Vaccination and/or Nonpharmaceutical Interventions Reduces the Rate at Which New Mutations Appear

Since higher values of reproductive number *R_T_* increase the likelihood that new mutations will arise within a specific lineage, controlling viral transmission should reduce the rate at which new variants with multiple mutations emerge. In our model, vaccination reduces the transmission rate by reducing the pool of susceptible individuals available for infection (Figure 2A). This time-independent reduction in *R_T_* decreases the rate at which new variants with two mutations emerge (Figure 2B,C).

Similarly, non-pharmaceutical interventions (NPIs) can reduce the SARS-CoV-2 transmission rate. However, many of these interventions reduce *R_T_* in a time-dependent manner, particularly if they are implemented in response to a COVID-19 outbreak. Therefore, we investigated the effect of NPIs on variant emergence by decreasing *R_T_* exponentially over time. In this scenario, transmission initially is high but slows to zero, as the intervention is implemented, resulting in a wave of SARS-CoV-2 infections. If the NPI is more effective (higher transmission reduction effect), this wave is shorter and involves fewer infections (Figure 3A). Limiting the overall spread of the single-mutant intermediate with NPIs reduces the likelihood that additional mutations will arise, even if the initial transmission rate *R_T_* of the intermediate is high (Figure 3B).

### 3.3. A Vaccine-Only Strategy for Controlling SARS-CoV-2 Spread Likely Cannot Prevent the Generation of New Immune-Evading Variants

Our simulations suggest that, if SARS-CoV-2 transmission is not controlled, new variants will continue to emerge relatively quickly. If these new variants can partially or fully evade the adaptive immune response to previous SARS-CoV-2 variants, sequential waves of transmission driven by new variants will occur. If these sequential outbreaks occur close together in time, it may be difficult to control and prevent these transmission waves by developing and deploying vaccine boosters against each variant.

To assess the feasibility of a vaccine-only strategy to control SARS-CoV-2 variant outbreaks in the US, we modeled viral transmission and vaccination using an SIR model. Initially, one individual is infected with a new SARS-CoV-2 variant, against which 30% of the population has effective immunity. At the time this variant appears, a vaccine that is effective against the variant is deployed and administered to a constant fraction of the susceptible population per day. No additional measures are implemented to slow the spread of the virus. If the vaccine is 100% effective against the initial variant, the US would have to vaccinate >1% of the population per day to reach a level of immunity in the population to prevent the emergence of a new variant if the virus is transmitting at the same rate as the ancestral strain (one conservative estimate of the ancestral *R*_0_ is 3.32) (Figure 4A). However, if the transmissibility of the virus is higher (as it is for the Delta and Omicron variants), the US would have to vaccinate more people per day to prevent new variants from emerging. For less effective vaccines, even faster deployment of a vaccine would be required (Figure 4B). These results suggest that efficient variant-specific vaccine deployment is unlikely to prevent new variants from repeatedly emerging and causing new waves of disease.

### 3.4. Vaccines Do Not Effectively Prevent the Generation and Selection of New Mutants within Individual Hosts

Another mechanism by which vaccination could slow SARS-CoV-2 evolution is by reducing the rate at which new beneficial mutants are produced within infected hosts. Our previous work has shown that viral load kinetics impact within-host evolution, with more viral replication leading to higher rates of variant generation [35]. Using our previously developed modeling strategy, we estimated viral load curves from PCR data [21] and simulated within-host evolution for vaccinated and unvaccinated individuals infected with Delta (Section 2). The published viral load data showed that viral expansion trajectories were similar between vaccinated and unvaccinated individuals, while the reduction in viral load during recovery was faster in vaccinated individuals (Figure 5A) [19,20,21]. We found that new beneficial viral variants with a single point mutation were generated and transmitted faster in vaccinated individuals than in unvaccinated individuals (Figure 5B,C). This is likely because the faster contraction of the viral population in vaccinated individuals imposes a greater selection pressure on the virus during later stages of infection, allowing beneficial mutants a greater advantage over wild-type virus. This advantage does not meaningfully increase the rate of double-mutant generation, which remains very infrequent in infections of typical length (Figure 5D).

We previously showed that generation of variants with more than one new mutation happens much faster in individuals with prolonged infections [35]. We found that mutation dynamics within individuals with prolonged SARS-CoV-2 infections are similar between vaccinated and unvaccinated individuals (Figure 6). This observation suggests that, unless vaccination affects the rate at which individuals develop longer infections, vaccinated individuals will generate new single- and multiple-mutation variants at a rate similar to that of unvaccinated individuals. Taken together, our results indicate that vaccination alone will not lower the rate of SARS-CoV-2 evolution enough to prevent the continued generation and selection for new variants.

## 4. Discussion

In this work, we have taken a first-principles approach rooted in the evolutionary theory to examine the impact of SARS-CoV-2-targeting vaccines on the rate of the variant emergence. Our results suggest that the critical determinant of the rate of SARS-CoV-2 variant emergence in a broadly vaccinated population is the impact of the vaccine on viral fitness. While the rate of evolution of resistance has been demonstrated in other settings to be proportional to the strength of the selection pressure [38,39], in the specific case of SARS-CoV-2 vaccines, it appears that vaccines that only exert a modest effect on transmission can still drive the evolution of resistance by facilitating the selection of immune-evasive mutants. Our approach and results are broadly applicable to other evolving pathogens for which vaccines are available. In particular, our model could be applied to the interplay between vaccine boosters and influenza evolution, especially since seasonal influenza is a good example of how pathogen evolution can be sustained despite frequent vaccine updates targeting specific variants.

In this work, we have examined two scenarios: the case of inter-host evolution and the case of intra-host evolution. In the case of inter-host evolution, we found that the generation of fitter mutants with multiple mutations is gated on the appearance of intermediate mutants with transmission rates high enough to permit spread within the population (Figure 1). To place this finding in context, recent waves of SARS-CoV-2 have had *R_T_s* ranging from approximately 1 to 3 [40]. High *R_T_s* values facilitate viral evolution by providing greater leeway for intermediate variants with reduced fitness to propagate. This finding illustrates a critical flaw in the public-health strategy during the current pandemic: relaxing restrictions in response to increased levels of vaccination within the population without implementing countervailing containment strategies speed up the emergence of vaccine-resistant variants. Notably, this outcome does not arise in the hypothetical case of broad population coverage with a vaccine that dramatically reduces transmission (Figure 2). Thus, in order for vaccination to provide a way out of the pandemic at this point, we need both vaccines that prevent infection and transmission and more widespread population coverage. Unfortunately, while many of available vaccines have been effective against severe disease and death, none effectively prevents infection (~80% effective against symptomatic infection and ~45–60% effective against asymptomatic infection) [41]. Therefore, the lack of vaccines that reduce infection risk remains a major barrier to curtailing SARS-CoV-2 evolution.

Our findings also underscore the importance of both behavioral and technical nonpharmaceutical interventions (NPIs) in supporting vaccines to slow viral evolution. NPIs complement vaccines in slowing viral evolution by limiting the transmission of novel vaccine-evading mutants (Figure 3). In the real world, vaccinal efficacy is degraded by two phenomena: the pharmacokinetic decay of neutralizing antibody titers and the rate of evolution of immune evasion by the virus (see [42] and Supplemental Text Section S8 in [43] for more details). We examined the hypothetical case of a vaccine whose efficacy was not degraded by immune-evading variants to identify the rate of vaccination required to prevent the generation of novel variants (Figure 4). Our findings suggest that, for a vaccine that is 100% effective at preventing transmission, boosters would be required once every 200 days to prevent variant emergence with an *R_T_* of 2.1. On the other hand, for a vaccine that is 70% effective, boosters would be required once every 66 days to prevent variant emergence with an *R_T_* of 2.1. For higher *R_T_s*, the use of vaccines to prevent variant emergence quickly becomes logistically unrealistic and may be immunologically implausible. Finally, we used a first-principles evolutionary modeling approach to examine the impact of vaccines on the emergence of immune-evading variants during intra-host evolution. Here, our findings suggest that vaccines are not expected to decrease the rate of generation of new variants within hosts during infections of typical (Figure 5) or prolonged duration (Figure 6). The latter will be particularly true for the accelerated evolution characteristic of long-term infection in immunocompromised individuals [35,44]. Taken together, our findings point to a minimal role for the current crop of vaccines in reducing the rate of emergence of novel immune-evading variants at this stage of the pandemic.

Popular misconceptions about viral evolution have repeatedly undermined the public-health response to this pandemic. Early in the pandemic, there was a widely held belief [45,46,47] that viral evolution would not pose a threat to the vaccines, as SARS-CoV-2 was perceived to be relatively stable [45,48,49]. In fact, it was easy to predict [28,50] that viral evolution would lead to rapid evolutionary escape, based on the extraordinary evolutionary plasticity of the spike protein [51,52,53]. Another commonly held perception has been that the virus will attenuate over time to become less virulent [48,54]. This is also demonstrably false—variants such as Alpha, Beta, Gamma, and Delta all had higher infection fatality rates (IFRs) than the ancestral strain (see Table S3 in [24]), and we have demonstrated in an earlier work that the IFR is not likely to be under strong selection pressure at this point [24], a point made by others as well [55]. Another common misconception is that T-cell activation by vaccines should provide stable protection against infection or severe disease. In fact, vaccinal protection against infection wanes quickly and is readily compromised by new viral variants [15,16,17]. Vaccinal protection against severe disease and hospitalization also wanes quickly [13] and is vulnerable to viral evolution, although to a more limited extent in the short term [14].

In this work, we draw attention to another common misconception about SARS-CoV-2 viral evolution, that is, widespread vaccination will help to slow it down. Our work suggests that the opposite can be true in certain settings—viral evolution can be facilitated by levels of vaccination in the population that are higher than those seen currently (at present, only around 33% (109 million) of the US population has received a first booster dose, and around 7% (23.5 million) has received a second booster dose [56]. Vaccines play a crucial role at present in limiting the acute morbidity and mortality burdens of COVID-19 and in preventing medical capacity from being overwhelmed. That said, relying solely on the current crop of vaccines to slow SARS-CoV-2 viral evolution is unlikely to be successful and may even make the problem worse.

Our work has a few key limitations. First, we assumed that new variants were created through the stepwise accumulation of point mutations, either with multiple mutations occurring within a single host or with mutations in the same viral lineage occurring in different hosts. Since many previous major SARS-CoV-2 lineages have arisen through the accumulation of point mutations [57,58], we did not include other mechanisms of genetic variation in our model, including recombination. Furthermore, we did not model superspreading, which has been shown to affect selection dynamics of SARS-CoV-2 [59,60]. More broadly, heterogeneity in disease transmission between individuals and between populations caused by differences in health status, geography, or demographics could affect our conclusions. While investigating the impact of this heterogeneity is not the focus of this study, additional modeling work could better resolve these effects. As another important caveat, we are not seeking to predict the short-term trajectory of the pandemic. Rather, the focus of this work is to point out that the currently available vaccines are not capable of solving the problem of rapid viral evolution of SARS-CoV-2 by themselves.

In summary, our work points to two unmet needs with the current public-health strategy for SARS-CoV-2. First, there is a need for better prophylaxis in order to bring SARS-CoV-2 under control. Vaccines (and antiviral prophylactics) that prevent infection and transmission are an urgent need at this point. Rapid viral evolution will continue to complicate the public-health response to the pandemic, will undermine new biomedical interventions as they become available [28], and may lead to unpredictable outcomes in the pandemic if left unchecked. Second, there is a need for a “stewardship” mindset in the use of existing and newly deployed vaccines and prophylactics. Public-health strategies must continue to support the increased use of NPIs, such as air quality improvement, testing-and-tracing, and masking, in tandem with encouraging vaccine uptake. Relying on vaccines alone to curb viral transmission places a focused evolutionary pressure on the viral spike protein, which will quickly degrade the utility of spike-targeting vaccines. At the current pace of evolution, the viral spike protein is evading the evolutionary pressure that we place on it faster than we can implement new interventions.

Three years into the pandemic, we face a virus that is evolving rapidly and is substantially more transmissible than it was at the outset. A clear-eyed understanding of the limitations of the tools at our disposal is crucial at this point for formulating public-health strategies. Our work highlights the importance of looking beyond the current crop of vaccines in finding measures to slow SARS-CoV-2 viral evolution.

## Figures and Tables

**Figure 1 vaccines-11-00853-f001:**
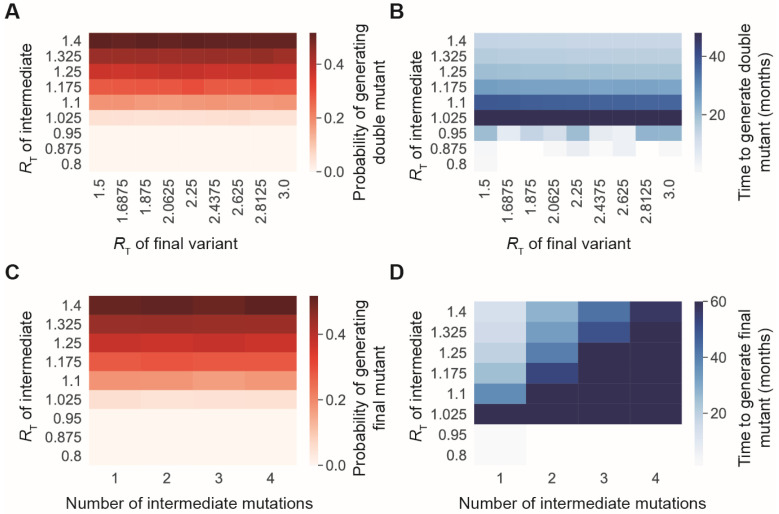
The transmission rate of intermediate SARS-CoV-2 mutants impacts the rate at which more transmissible variants with multiple mutations appear. (**A**) Probabilities of generating a double-mutant lineage that establishes in the population before the extinction of the single mutant, for different *R_T_* values of the single mutant (*y*-axis) and the double mutant (*x*-axis) *R_T_*. (**B**) Time to generate a double mutant variant after the initial mutation occurs in one host, given that a double mutant eventually arises. This was plotted for different *R_t_* values of the single mutant (*y*-axis) and the double mutant (*x*-axis) *R_T_*. (**C**) Probability of generating a variant with multiple mutations from a specific single-mutant lineage before its extinction, for different values of the transmissibility of the intermediate variants (*y*-axis) and the number of intermediate mutations required before the final variant can be generated (*x*-axis). (**D**) Time to generate a multiple-mutant variant after the initial mutation occurs in one host, given that the final variant eventually arises from the original single mutant. This was plotted for different values of the transmissibility of the intermediate variants (*y*-axis) and the number of intermediate mutations required before the final variant can be generated (*x*-axis). For all panels, *n* = 10,000 simulation was run.

**Figure 2 vaccines-11-00853-f002:**
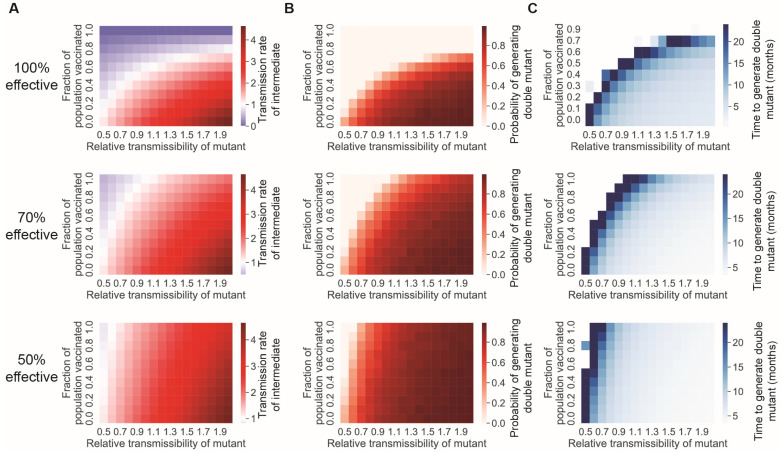
Vaccination can lower the rate at which new variants appear by lowering the effective transmission rate. (**A**) The effective transmission rates of single mutants, given different coverage fractions of a completely effective vaccine, for a 100%, 70%, or 50% effective vaccine against transmission. The *x*-axis denotes the transmissibility of the single-mutant variant, relative to the original WT transmission rate *R_T_* of 3.32. (**B**) Probability of generating a double-mutant lineage before the extinction of the single mutant. (**C**) Time to generate a double-mutant variant after the initial mutation occurs in one host, given that a double mutant eventually arises. For all panels, *n* = 10,000 simulation was run.

**Figure 3 vaccines-11-00853-f003:**
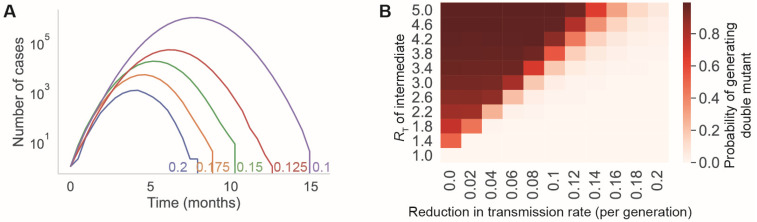
Limiting the transmission of a new SARS-CoV-2 single mutant reduces the risk of generating more-transmissible variants. (**A**) Example infection prevalence trajectories with different transmission reduction rates (colors; parameter values labeled on the plot). (**B**) Probabilities of generating a double-mutant lineage before the extinction of the single mutant, for different values of the single-mutant transmissibility (*y*-axis) and the transmission reduction rate (*x*-axis). Each parameter set had *n* = 10,000 simulation runs.

**Figure 4 vaccines-11-00853-f004:**
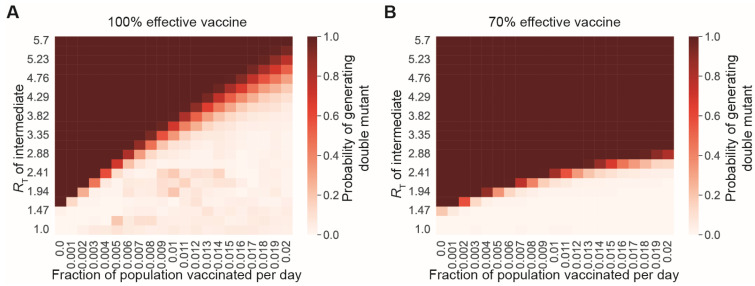
Vaccination alone will not prevent the generation of new variants, if transmission rates are high. (**A**,**B**) Probabilities of generating a double-mutant lineage before the extinction of the single mutant, for different values of the single-mutant transmissibility (*y*-axis) and the fraction of the unvaccinated and the nonimmune population vaccinated per day (*x*-axis). The vaccine is either completely effective in preventing infection from single-mutant variants (**A**) or 70% effective in preventing infection (**B**). For both panels, *n* = 1000 simulation was run, and the total population size was 300 million individuals.

**Figure 5 vaccines-11-00853-f005:**
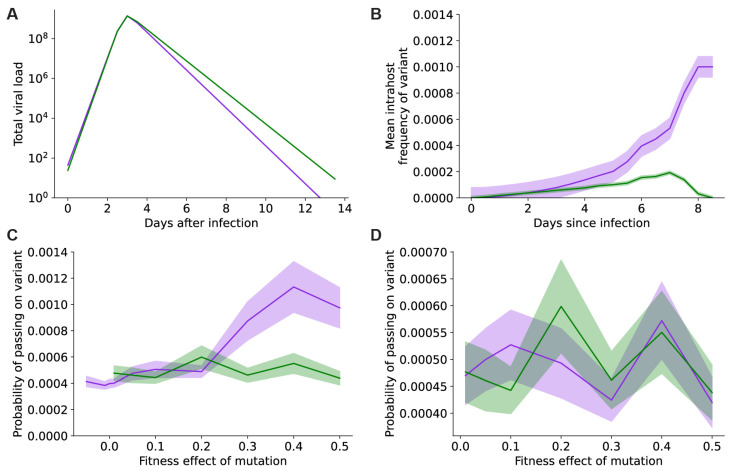
Vaccination does not decrease the rate of generation of new variants within hosts during typical length infections. (**A**) Viral load profiles for vaccinated (purple) and unvaccinated (green) individuals infected with the Delta SARS-CoV-2 variant. (**B**) Frequencies of an advantageous viral variant with a new single mutation (20% fitness benefit over the wild-type) within a vaccinated (purple) or unvaccinated (green) infected individual over a typical-length SARS-CoV-2 infection. (**C**) Probability of generating and transmitting an advantageous viral variant with a new single mutation with a 20% fitness benefit over the wild-type from a host that was vaccinated (purple) or unvaccinated (green). (**D**) Probabilities of generating and transmitting an advantageous viral variant with two new mutations with a 50% fitness benefit over the wild-type from a host that was vaccinated (purple) or unvaccinated (green). In (**B**–**D**), shaded regions represent 95% binomial confidence intervals.

**Figure 6 vaccines-11-00853-f006:**
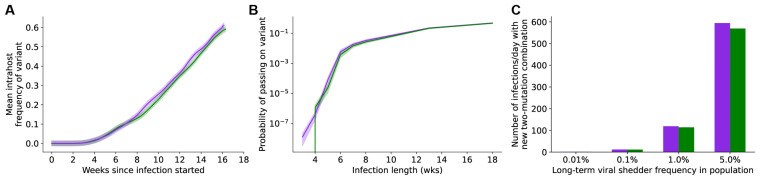
Vaccination does not decrease the rate of generation of new variants with multiple mutations within hosts with prolonged SARS-CoV-2 infections. (**A**) Frequencies of an advantageous viral variant with two new point mutations (final variant has a 50% fitness benefit over the wild-type) within a vaccinated (purple) or unvaccinated (green) infected individual over a prolonged SARS-CoV-2 infection. (**B**) Probabilities of generating and transmitting an advantageous viral variant with two new point mutations (final variant has a 50% fitness benefit over the wild-type) within a vaccinated (purple) or unvaccinated (green) infected individual over a prolonged SARS-CoV-2 infection. In (**A**,**B**), shaded regions represent 95% binomial confidence intervals. (**C**) Population-level rates of generating new two-mutant SARS-CoV-2 variants with a 50% fitness benefit assuming different frequencies of long-term SARS-CoV-2 infections in the population. All infected individuals were assumed to either be vaccinated (purple) or unvaccinated (green).

## Data Availability

No new experimental data were generated in this simulation study. The branching process and SIR simulations were implemented in Python, and codes for running these simulations and plotting the results are available in a Jupyter notebook at https://github.com/dvanegeren/covid-var-emerge (accessed on 11 April 2023).

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
