# Peer review of "Vaccines Alone Cannot Slow the Evolution of SARS-CoV-2"

_vaccines, 2023, doi:10.3390/vaccines11040853_

Round 1
Reviewer 1 Report
REVIEW REPORT
Title: Vaccines alone cannot slow the evolution of SARS-CoV-2
Article type: Article
ID: vaccines-2314159
Date: 19.03.2023
General comments:
The authors presented their model with subsequent interpretations regarding overall merit and its applicability. It is well written. The authors stressed the need for a combination of vaccines (+ antiviral prophylactics) as well as NPIs to tackle the problem of the pandemic. The topic in the form of transmission dynamics has been studied in the past. For example, a mathematical model has been proposed to determine whether or not a hypothetical imperfect vaccine can lead to the elimination of COVID-19 in the United States.
The authors (Iboi et al.) concluded that for an anti-COVID-19 vaccine with an assumed protective efficacy of 80%, at least 82% of the susceptible US population would need to be vaccinated to achieve the herd immunity threshold. What is more, combinations of strategies (masks + vaccination) significantly reduce the level of the vaccine-induced herd immunity threshold needed to eliminate the pandemic in the US ((72% if half of the US population regularly wears face masks in public (the threshold decreases to 46% if everyone wears a face mask)).
It would be interesting to know whether it is possible to include another variable in the model, namely the geographical differences (geographical disproportion in the occurrence of consequential variant outbreaks, e.g., A simulation of geographic distribution for the emergence of consequential SARS-CoV-2 variant lineages; Akaishi et al.; or The dynamics of novel corona virus disease via stochastic epidemiological model with vaccination) as well as sociodemographic differences (e.g., Socio-demographic and health factors drive the epidemic progression and should guide vaccination strategies for best COVID-19 containment; Markovič et al.).
With best regards.
Author Response
The reviewer is correct to point out that there is variation in epidemiological dynamics between different geographical regions and demographic groups that could modulate how vaccination affects viral evolution. Much of this variation can be represented by differences in contact rate which affects the overall transmission rate. To address this, we swept across a range of possible viral transmission rates (here, parameterized as RT) in Figs. 1, 2, and 4, which could represent not just intrinsic differences between strains in transmissibility, but also differences between population-level transmission dynamics. However, there are additional sources of interindividual heterogeneity in transmission parameters (e.g., superspreading, variation in immune response) that are not well captured by a population-level parameter such as RT. While investigation of these sources of variation are not the focus of this study and would require extensively reformulating our computational model, we have explored some of these effects in other work (e.g., Stoddard M, …, Chakravarty A. medRxiv 2023) and are continuing to investigate the effects of superspreading on viral dynamics in separate follow-on studies. We have edited the relevant passage in the Discussion (lines 405-408) highlighting this limitation of our current work.
Reviewer 2 Report
The manuscript by Debra Van Egeren et al described a study on evaluation of the impact of vaccines to the evolution of SARS-CoV-2. The findings are novel and of interest. My major concern is how to quantify the impact of a vaccine? As there are many kinds of vaccines against SARS-CoV-2. Is there any difference among the vaccines?
Author Response
We thank the reviewer for their interest in our findings. The reviewer is correct to point out differences between different SARS-CoV-2 vaccines. These differences are often quantified as the efficacy or protection against infection, severe disease, or death. A recent meta-analysis (Yang Z*, Jiang Y*, …, Tang J. Lancet Microbe 2023) showed that SARS-CoV-2 vaccines have variable efficacies, with mRNA vaccines in particular showing better and more durable responses against infection. We modeled vaccine efficacy in a population with a particular fraction of vaccinated people as a reduction in overall transmissibility, as described in the Materials and Methods (lines 103-111), and ran simulations using different vaccine efficacies (50-100% effective against infection) in Figs. 2 and 4 to illustrate the impact of vaccine efficacy on our conclusions. We have added some information on the measured vaccine efficacies against infection to the Discussion (lines 345-349) to help readers better interpret our results.
Reviewer 3 Report
The aim of the Authors is to evaluate a stochastic computational model of the transmission of SARS-CoV2 and related variants.
In order to better analyze the dinamics of the transmission, the AA simulated “the emergence of new SARS-Cov2 variants whit one or more mutations as a stochastic branching process with discrete generations”, also considering “the immunity due to previous infection or vaccination”, and with a first hypothesis of constant RT over time.
Then they applied this SIR model to evaluate the effectiveness of the vaccination in avoiding or reducing the creation of new variants.
Finally, they performed a simulation of intrahost mutation dynamics.The results obtained are consistent with epidemiological studies and this supports the validity of the study.
Altogether, the results obtained are very interesting and this model appears very original and likely to be also applied to other epidemics or pandemics caused by respiratory viruses.
Minor comments:
A further contribution could be done by performing separate assessments according to the type of vaccination, the number of doses and taking into account facing climatics extremes (eg Switzerlang, Kenya, etc.) and different lifestyles (e.g. urban or rural populations, African or European people, etc).
A comparative analysis with other pandemic or epidemic emergencies from mutant respiratory viruses might be useful.
Author Response
We appreciate the reviewer’s positive comments on the potential impact of our work. We agree that population-level heterogeneity in environment or demographics can lead to relevant differences in disease transmission dynamics. Some of this heterogeneity is reflected in differences in the contact rate, which affects the effective transmission rate RT. To address variation and/or uncertainty in this parameter, we have swept across a range of values in Figs. 1, 2, and 4. We have also addressed variation in vaccine type by simulating different values for the vaccine efficacy in Figs. 2 and 4. However, to better clarify how our simulation results relate to different real-world vaccine types, we have added information on measured vaccine effects in the Discussion (lines 345-349).
We also agree that our methods and results may be applicable to outbreaks caused by other pathogens. The best such example of a respiratory virus for which the interplay between evolution and vaccination is highly important is influenza. In fact, seasonal influenza represents an example of a respiratory virus whose evolution is not adequately constrained by annual vaccine boosters, leading to tens of millions of yearly cases in the US alone. This is despite our ability to forecast dominant strains using geographical variation in influenza seasonality and design appropriately targeted vaccines. While fully modeling influenza virus evolution (while taking into account influenza’s particular evolutionary constraints, as reviewed in Petrova VN and Russell CA. Nat Rev Microbiol 2017) is outside the scope of our study, we have added text (lines 326-330) stating that our approach could be broadly applicable to other pathogens to address the reviewer’s intriguing suggestion.
Reviewer 4 Report
The paper entitled « Vaccines alone cannot slow the evolution of SARS-CoV-2” by Debra van Egeren et al presents modeling analysis of SARS-CoV-2 infection in presence or not of vaccination regimen and the emergence of immune-evading variants. The work is clearly presented and the data are interestingly discussed and the conclusions are in agreement with the data. The main conclusion of the paper is to plead for better prophylaxis in order to bring the virus under control among the human population but also to maintain, with the vaccination, the use of non-pharmaceutical intervention such as masking, testing-tracing and air quality checking and improvement. The analysis was undertaken very accurately and takes into account the previously reported data on this subject. It offers for the readers a major opportunity for further scientific discussions on this very important subject.
Author Response
We thank the reviewer for their accurate summary of our findings and their positive comments on our work.